# Epigenetics as a Therapeutic Target in Osteoarthritis

**DOI:** 10.3390/ph16020156

**Published:** 2023-01-21

**Authors:** Carmen Núñez-Carro, Margarita Blanco-Blanco, Karla Mariuxi Villagrán-Andrade, Francisco J. Blanco, María C. de Andrés

**Affiliations:** 1Unidad de Epigenética, Grupo de Investigación en Reumatología (GIR), Instituto de Investigación Biomédica de A Coruña (INIBIC), Complexo Hospitalario Universitario, de A Coruña (CHUAC), Sergas, 15006 A Coruña, Spain; 2Grupo de Investigación en Reumatología y Salud, Departamento de Fisioterapia, Medicina y Ciencias Biomédicas, Facultad de Fisioterapia, Campus de Oza, Universidade da Coruña (UDC), 15008 A Coruña, Spain

**Keywords:** epigenetics, osteoarthritis, DNA methylation, histone methylation, histone acetylation, miRNA, circRNA, lncRNA

## Abstract

Osteoarthritis (OA) is a heterogenous, complex disease affecting the integrity of diarthrodial joints that, despite its high prevalence worldwide, lacks effective treatment. In recent years it has been discovered that epigenetics may play an important role in OA. Our objective is to review the current knowledge of the three classical epigenetic mechanisms—DNA methylation, histone post-translational modifications (PTMs), and non-coding RNA (ncRNA) modifications, including microRNAs (miRNAs), circular RNAs (circRNAs), and long non-coding RNAs (lncRNAs)—in relation to the pathogenesis of OA and focusing on articular cartilage. The search for updated literature was carried out in the PubMed database. Evidence shows that dysregulation of numerous essential cartilage molecules is caused by aberrant epigenetic regulatory mechanisms, and it contributes to the development and progression of OA. This offers the opportunity to consider new candidates as therapeutic targets with the potential to attenuate OA or to be used as novel biomarkers of the disease.

## 1. Introduction: The Concept of Epigenetics and Its Relevance in the Development of Diseases

The term ‘epigenetics’ was first coined by Conrad Waddington in 1942 [1], but the discovery of its relevance in a myriad of biological processes did not occur until the last few decades. Since then, the concept has evolved from being considered as ‘changes in phenotype without changes in genotype’ to a more specific and dynamic approach. Currently, ‘epigenetics’ refers to heritable or dynamic changes without altering the primary DNA sequence, adapting the conformation of chromatin to regulate gene expression [2].

The epigenome is susceptible to heritable and/or dynamic changes in the early stages of development, during the prenatal and early postnatal periods, e.g., transgenerational effects, maternal care, breastfeeding, or maternal diet. These changes in the epigenetic landscape, especially at the methylome level, may predispose to certain diseases in adulthood [3]. For example, cancer, obesity, metabolic syndrome, cardiovascular disease, low birth weight, developmental diseases, mental illness, or diabetes may be caused by alterations in the epigenome [4]. In an adult individual, the epigenome is more stable, although it can also be altered by environmental influences and lifestyle [3]. Epigenetics is also important at the end of our lives, as it is closely related to age and age-related diseases. DNA methylation and demethylation, histone modifications and non-coding RNA (ncRNA) modifications are the main epigenetic mechanisms involved in the regulation of gene expression. These basic epigenetic mechanisms can be altered as a result of the aging process, in which aberrant patterns of DNA methylation, accumulation of histone variants, imbalance of histone modifications, loss of canonical histones and heterochromatin, and abnormal activity of microRNAs on the transcriptome may occur [5]. Therefore, considering that osteoarthritis (OA) is one of the main causes of disability worldwide and that its prevalence is expected to continue to increase in the coming years [6], it is necessary to expand knowledge about the mechanisms on which OA operates and thus be able to find new therapeutic targets to delay or halt its progression. Evidence suggests that different environmental factors, including joint trauma, oxidative and inflammatory stress, aging, diet, metabolic disorders, or genetics (both nuclear and mitochondrial), may induce epigenetic changes that could favour the development of different phenotypes of OA [7]. Thus, elucidating the role of epigenetics in OA has become a current priority. In accordance with this, the present review will analyse the most recent literature to highlight the importance of epigenetics at the therapeutic level in OA.

## 2. Literature Search Method

The literature was systematically reviewed across the PubMed database (https://pubmed.ncbi.nlm.nih.gov/, accessed on 12 January 2023). Studies were selected within the field of epigenetics in OA by performing searches in blocks corresponding to the different sections included in the review, i.e., ‘epigenetics’ and ‘osteoarthritis’, ‘osteoarthritis’ and ‘DNA methylation’, ‘osteoarthritis’ and ‘histone’, ‘osteoarthritis’ and ‘histone methylation’, ‘osteoarthritis’ and ‘histone acetylation’, ‘osteoarthritis’ and ‘PTM’, ‘osteoarthritis’ and ‘miRNA’, ‘osteoarthritis’ and ‘circRNA’, and ‘osteoarthritis’ and ‘lncRNA’. In certain sections we narrowed the search by adding a specific enzyme as a parameter, for example, ‘osteoarthritis’ and ‘SIRT1’ or ‘osteoarthritis’ and ‘HDAC1’. We focused on articles written in English and published in the last five years, except in some cases where information was very scarce. We considered studies performed in humans as well as in cellular and animal models. A total of 161 publications were consulted for this review.

## 3. Osteoarthritis (OA): A Whole Joint Disease

OA is a chronic, heterogeneous disease and is currently the most common form of arthritis [8]. Traditionally, OA has been considered a degenerative disease affecting articular cartilage, but now it is firmly established that OA is an inflammatory pathology that affects the whole joint, including cartilage, bone, and synovium. Some pathological changes caused by OA are cartilage degradation, thickening of the subchondral bone, osteophyte formation, and synovial inflammation, among others [9]. The present review focuses on the epigenetics of the osteoarthritic articular cartilage, which is a connective tissue lacking blood vessels, lymphatic vessels, and nerves. The only cell type in articular cartilage is the chondrocytes, highly specialised cells that follow a scattered distribution across the extracellular matrix (ECM). The ECM is composed of water, collagen, proteoglycans (PGs), and different proteins. Together, these elements provide a suitable surface to allow joint movement, minimising friction [10].

The components of the joint are altered in quantity and in their elastic properties in OA, and this structural and functional disorganisation of cartilage leads to its progressive destruction [9]. During aging, a state of chronic low levels of inflammation occurs, which is termed ‘inflamm-aging’ [11]. This process appears to contribute, in part, to the imbalance between chondrocyte anabolism and catabolism [12]. In general, this happens when chondrocytes respond to an injury produced in the joint, since they are cells capable of detecting mechanical stimuli [13]. When this occurs, there is an increase in the expression of certain molecules that are closely related to innate immunity (tab 1) [14]. Prominent among these is interleukin 1β (IL-1β), which is overexpressed in OA cartilage and synovial tissues [15]. This cytokine is able to increase the expression of ECM matrix metalloproteinases (MMPs) and aggrecanases, whose action promotes ECM destruction and inhibits the synthesis of its major components, such as type II collagen (COL II) and PGs [16]. Another essential factor is tumour necrosis factor α (TNF-α), a growth factor that directs the inflammatory signalling cascade inducing the production of proinflammatory and catabolic molecules [17]. Transforming growth factor β (TGF-β) is also involved in OA, but its role is controversial due to its dual nature. TGF-β acts in a complex and contextual manner, and can activate signalling pathways that promote anabolism as well as those that favour catabolism. For example, TGF-β can activate the ALK5-SMAD2/3 pathway to induce an anabolic response that induces cartilage ECM turnover, but it can also activate the ALK1-SMAD1/5/8 cascade that promotes the expression of catabolic molecules responsible for cartilage destruction [18]. If these proinflammatory cytokines are detected by chondrocytes, the expression of the enzyme inducible nitric oxide synthase (iNOS) is activated (Figure 1). iNOS, in response, synthesises nitric oxide (NO), a gas that acts as a defence agent due to its oxidative power and toxicity [19]. However, if NO production is excessive and the cells′ ability to detoxify reactive oxygen species (ROS) is weakened (as it happens in OA), the cell will undergo oxidative stress. Some effects of this, such as degradation of the plasma membrane or nucleic acids, are fatal to the cell and ultimately cause cell death. In addition, degradation of ECM components, such as collagen or PGs, is also frequent. Consequently, the amount of these molecules increases in the synovial fluid (SF). All this results in a vicious circle in the cellular environment, as the ROS content of the apoptotic cells will exert stress on the surrounding cells, which will also form new ROS (Figure 1) [20]. This is the typical scenario of OA cartilage and is related to chondrosenescence. This process is mainly characterised by a decrease in the proliferative capacity of chondrocytes and in the synthesis of antioxidant enzymes [19]. However, the production of proinflammatory mediators and ECM-degrading enzymes does not decrease in step with cell proliferation [20]. In addition, the remaining chondrocytes attempt to compensate for the loss of structural molecules in the ECM by replicating and forming clusters of approximately 50 cells known as “chondrocyte clones” [21]. These cells inherit by mitosis a repertoire of genes whose epigenetic pattern is aberrant. As a result, they neo-express a battery of degradative enzymes that healthy chondrocytes do not express due to epigenetic silencing [22].

Numerous studies that will be reviewed in further sections support that the development of this aberrant inflammatory activity is caused, in part, by a dysregulation of proinflammatory molecules and pathways at the epigenetic level. This puts the spotlight of research on epigenetics to find new biomarkers or therapeutic targets against OA.

## 4. DNA Methylation and Demethylation in OA

DNA methylation is mainly catalysed by DNA methyltransferases 1, 3A, and 3B (DNMT1, DNMT3A, DNMT3B) and consists of the addition of a methyl group (-CH_3_) to cytosine (C) nucleotides located 5′ with respect to a guanine (G), resulting in 5-methylcytosine (5-mC) production. DNMT3A and DNMT3B establish de novo methylation patterns, while DNMT1 is in charge of methylation maintenance [23]. The reverse process or DNA demethylation occurs by ten-eleven translocation enzymes 1, 2, and 3 (TET1, TET2, TET3), which form 5-hydroxymethylcytosine (5-hmC) [16]. This type of epigenetic modification is especially common in cytosines of CpG dinucleotide sequences or CpG islands, which are abundant in promoter regions of genes [24]. Generally, CpG methylation correlates with transcriptional repression, as it hinders the access of transcriptional machinery to the DNA sequence [25].

DNA methylation plays a crucial role in OA pathogenesis (Figure 2). Imagawa et al. demonstrated that the *COL9A1* gene, which encodes an essential ECM component, presented six hypermethylated CpG sites in OA patients. This, in turn, induces the downregulation of *COL9A1* expression, promoting the loss of cartilage integrity [26]. The enzymes responsible for DNA methylation, the DNMTs, also act as critical regulators of epigenetic regulation in OA. Zhu et al. investigated the decrease of peroxisome proliferator-activated receptor gamma (PPARγ) in human and murine OA [27]. They found that DNMT1 and DNMT3A (which are elevated in OA cartilage) hypermethylated the *PPARG* promoter, hence suppressing PPARγ expression. This translated into the exacerbation of OA. Thus, the authors investigated the effect of the pharmacological DNA demethylating agent 5-Aza-2’-deoxycytidine and discovered that it restored PPARγ expression, conferring chondroprotection by alleviating oxidative stress and inflammatory response. Moreover, inhibition of Dnmt1 and Dnmt3a in a destabilisation of the medial meniscus (DMM) model protected murine articular cartilage against OA, making these enzymes therapeutically interesting [27]. Shen and collaborators demonstrated that DNMT3B is highly expressed in healthy human cartilage, yet its expression decreases in OA human chondrocytes as well as in OA mouse models. Furthermore, *Dnmt3b* knockdown triggers, in part, the early onset and progression of an OA with accelerated tricarboxylic acid (TCA) cycle and abnormally increased mitochondrial respiration [28]. Meanwhile, *Dnmt3b* gain of function results in increased *Col2a1* levels and reduced Runt-related transcription factor 2 (*Runx2*) and *Mmp13* expression, thereby exerting a chondroprotective effect [28]. More recently, the same authors decided to focus their efforts on the study of the downstream targets of Dnmt3b in order to find more suitable candidates to be therapeutic targets, since Dnmt3b acts all across the genome [29]. Interestingly, they discovered that 4-aminobutyrate aminotransferase (Abat) (which catalyses γ-aminobutyric acid (GABA) to succinate, an essential metabolite in the TCA cycle [30]) is a downstream target of Dnmt3b in chondrocytes and its inhibition in vivo attenuates murine OA progression [29]. 

TET enzymes, especially TET1, also contribute significantly to the pathogenesis of OA. In fact, the complete absence of *Tet1* protects the joint in a murine model of OA by preventing cartilage surface destruction and osteophyte formation [31]. Since TET1 activity is inhibited in response to inflammatory factors in human OA chondrocytes, a decrease in 5-hmC levels would be expected, but they are found to be increased [32]. This elevated amount of 5-hmC in promoter regions of *MMP*-encoding genes results in overexpression of cartilage-degrading MMPs [33]. The decrease in TET1 and increase in 5-hmC could be independent events occurring in OA, or prolonged exposure to inflammatory cytokines and, thus, prolonged insufficiency of TET1 may be required for 5-hmC to accumulate in cells. Considering that TET enzyme homeostasis in OA appears to be altered, it is likely that the elevated rate of 5-mC–5-hmC conversion is not solely dependent on the catalytic activity of TET enzymes [33]. Further research is required to elucidate the contribution of TET enzymes to OA disease.

Several studies support that hypomethylation is a key process in the altered synthesis of some OA-related molecules (Figure 2). Hypomethylation of CpG sites of genes encoding for MMPs and aggrecanases causes an increased expression of these enzymes in OA cartilage [34,35,36]. On the other hand, it has been proven that hypomethylation in a *NOS2* enhancer region leads to iNOS overexpression in OA, as binding sites for NF-κB, a key transcription factor in inflammation, become more accessible [37]. In the same way, demethylation of a promoter region of interleukin 8 (*IL8*) causes an increased expression of this proinflammatory chemokine in OA chondrocytes [38], which has been associated with ECM PG loss, MMP production, chondrocyte hypertrophy, and apoptosis, as well as with the entry of immune system cells into synovial tissue [39]. Likewise, interleukin 6 (*IL6*) promoter exhibits marked hypomethylation in the context of OA that leads to IL-6 overexpression in synovial fibroblasts and SF [40]. Another study detected hypomethylation in CpG sites of *Cebpa, Cdk2*, *Bak1,* and *Fas* genes in rats with knee OA (which are mainly related to cell differentiation, proliferation, and apoptosis), thereby increasing their expression [41]. Equally, the increased production of RUNX2 occurs because specific CpG sites within *RUNX2* P1 promoter are hypomethylated. Accordingly, high levels of RUNX2 mediate *MMP13* enhanced transcription, which encodes a major ECM disruptor enzyme that is typically overexpressed in OA [42].

Over the last few years, research in the field of DNA methylation has turned to genome-wide studies to identify differentially methylated genes in OA. Zhao et al. identified 84 genes with different methylation patterns, among which 45 genes differed significantly between OA and control chondrocytes. Of these 45 genes, 23 showed hypermethylation and the remaining 22 showed hypomethylation. Going deeper into the study, the authors demonstrated that TNF receptor-associated factor 1 (*TRAF1*), connective tissue growth factor (*CTGF*) and C-X3-C motif chemokine ligand 1 (*CX3CL1*) mRNA expression levels are considerably upregulated in OA chondrocytes as a consequence of their genes’ demethylation [43]. Although *TRAF1*, *CTGF,* and *CX3CL1* have not been widely studied yet, these results suggest that they contribute to OA pathogenesis. Wang and colleagues conducted a genome-wide DNA methylation profiling of articular cartilage from Kashin–Beck disease (KBD) patients, which is an endemic OA from certain regions of Asia [44]. Comparing OA and KBD patients, they found 367 differentially methylated positions corresponding to 182 genes that overlapped between both diseases. Additionally, all these genes presented matching methylation directions in OA and KBD compared to control patients’ methylation status [45].

Genome-wide studies have elucidated the existence of a mechanistic interplay between genetics and epigenetics in OA [46]. Fernández-Tajes et al. examined the genome-wide DNA methylation profile of articular human chondrocytes in order to identify characteristic profiles of DNA methylation in OA. They detected a tight cluster of patients with OA within a cohort comprised of 25 patients with OA and 20 healthy controls. In this subgroup of patients, 1357 differentially methylated probes were detected with respect to other OA patients. Moreover, 450 genes were differentially expressed in the patients included in this cluster, and this translated into an increased inflammatory response that could be epigenetically regulated [47]. Recently, some single nucleotide polymorphisms (SNPs) have been identified as markers of OA genetic risk loci. For instance, rs11583641 is located in *COLGALT2*, which encodes a glycosyltransferase responsible for post-translational modification of ECM collagen. This SNP alters DNA methylation at a gene enhancer, influencing *COLGALT2* expression. Thus, the post-translational modification of collagen may be affected and this, in turn, could have relevance in the pathogenesis of OA [48]. Another OA risk SNP of significant relevance is rs75621460. Located at a *TGFB1* enhancer region, it decreases the expression of the well-studied anabolic factor *TGFB1*, compromising cartilage integrity. Therefore, the identification of *TGFB1* as an OA risk gene could enable the search for new specialised therapeutic interventions in subgroups of patients with this SNP [49].

## 5. Histone Modifications in OA

Histones are highly basic proteins that, together with DNA, constitute the fundamental unit of eukaryotic chromatin: the nucleosome. DNA is wrapped around a histone octamer comprised of two copies of H2A, H2B, H3, and H4, also known as core histones [50]. Located between two nucleosomes is histone H1, a linker protein that enables the correct organisation of higher order chromatin structures and plays a role in nucleosome spacing across the genome [51]. 

Histones undergo highly dynamic post-translational modifications (PTMs), such as acetylation, methylation, phosphorylation, ubiquitylation, and many others. PTMs can condition the structure of chromatin, causing it to acquire a looser organisation, known as euchromatin, or a more compact one, known as heterochromatin [52]. The conformational state of chromatin is determined by the crosstalk between DNA and histones. High levels of acetylated histones and hypomethylated DNA regions are closely related to euchromatin, whereas the absence of acetylation in histones and DNA hypermethylation are features of heterochromatin [53]. 

Evidence that histone PTMs may play a significant role in OA has grown over the past few years. However, DNA–histones interaction is still not well understood and requires further research. Particularly in the case of OA, the role of histones in the pathology onset and development remains largely unknown. This is partly due to the difficulty of obtaining these proteins from articular cartilage, a tissue that is characterised by its structural complexity and low cellularity. Such features complicate certain in vitro procedures, such as histone extraction. Because of this, a protocol for extracting histones from human articular cartilage has recently been published in the hope of assisting in the understanding of histone influence in the pathogenesis of OA [54].

### 5.1. Histone Methylation and Demethylation in OA

As in the case of DNA, histone methylation–demethylation is one of the most studied types of histone PTMs, and it is catalysed by histone methyltransferases (HMTs) and histone demethylases (HDMTs), respectively. Histone methylation can be related to either transcription silencing or activation depending on which residue of lysine (K) or arginine (R) in the histone tail is modified and how many methyl groups are added [55]. Although histone PTMs with relevance in OA have not yet been fully characterised, there are examples of their possible involvement as epigenetic regulators in the pathology. 

Trimethylation of H3K9 (H3K9me3) and H3K27 (H3K27me3) has been associated with transcriptional repression [56]. Under OA conditions, an elevated level of H3K9me3 and H3K27me3 can be observed in the *SOX9* gene promoter, which encodes a pivotal transcription factor in chondrogenesis and determination of chondrocyte identity. This suggests that the typical low SOX-9 expression in OA at both gene and protein levels could be a result of changes in the epigenome [57]. Another study showed a global reduction of H3K9 methylation at various stages (H3K9me1, H3K9me2, and H3K9me3) in damaged articular cartilage of the temporomandibular joint (TMJ) in aged mice [58]. An in vitro inhibition of a H3K9-specific methyltransferase in the ATDC5 chondrogenic cell line revealed an increased apoptosis, upregulation of ECM-degrading enzymes (*Mmp1* and *Mmp3*), and downregulation of anabolic factors (*Sox9* and *Col2a1*), supressing cell proliferation and negatively regulating chondrocyte homeostasis [58]. Moreover, Maki and collaborators demonstrated that the application of hydrostatic pressure (HP) on ATDC5 chondrocytes mechanically regulates chromatin conformation and chondrocyte differentiation. HP induces a reversible shrinkage of the cell nucleus and chromatin loosening by a reduction in H3K9me3 and a parallel increase in H3K27me3 in heterochromatin regions, attenuating the expression of hypertrophy-related genes. In addition, they promote the entry of chondrocytes into a quiescent state that could prevent their excessive maturation to the hypertrophied state they exhibit in advanced stages of OA [59]. 

In contrast, H3K4me3 is involved in transcriptional activation and it is widely distributed across the CpG islands [56]. The fact that H3K4me3 has been reported to be sensitive to ROS levels [60] has sparked interest in the field of OA, since ROS play a major role in the disease. Dexamethasone (Dex) is a widely used drug that alleviates OA symptoms, since it exerts anti-inflammatory and anticatabolic effect on chondrocytes [61]. Conversely, Li et al. conducted an intraarticular (IA) Dex rat model and observed that Dex could damage cartilage by decreasing anabolic ECM genes *Col2a1* and aggrecan (*Acan*), increasing *Mmp13* expression and promoting chondrosenescence. This divergence between both studies could be attributed to experimental differences. Moreover, mitochondrial dysfunction and ROS production were increased in human chondrocytes stimulated with 1 µM Dex [62]. High levels of ROS decrease carboxylate transporter 4 expression (MCT4), which is implicated in maintaining a normal intracellular pH in the chondrocyte. This reduction is, in fact, a consequence of a reduction in H3K4me3 in the *Mct4* gene [62].

In any case, histone methylation-related PTMs are the product of the enzymes that catalyse them. Among these, there are also several examples that illustrate that they may be dysregulated in OA (Figure 3). Prominent among HMTs is the enhancer of zeste homolog 2 (EZH2), which has gained importance in the OA field in recent years. Accumulating evidence supports the role of EZH2 as a critical mediator of cartilage destruction, since it is strongly upregulated in OA [63]. EZH2 possesses specific H3K27 methyltransferase activity, through which it is capable of inducing chondrocyte hypertrophy and promoting the expression of catabolic factors (COL type X, Indian Hedgehog, MMP-1, MMP-3, MMP-13, ADAMTS-4, and ADAMTS-5) through the Wnt/β-catenin signalling pathway [63,64] (well-known to induce cartilage destruction in OA [65]). Additionally, EZH2 overexpression in IL-1β-stimulated chondrocytes increases inflammation-related molecules production, such as prostaglandin (PGE2), NO, and IL-6 [64]. Nonetheless, treatment with EPZ-6438 (an EZH2 inhibitor) in periodontal ligament stem cells (PDLSCs) lacking lysine demethylase 6A (KDM6A) restored their chondrogenic potential by decreasing H3K27me3 [66]. Additionally, the use of more pharmacological inhibitors of EZH2-mediated hypermethylation have been proven to delay the development of OA [63,64], making EZH2 a promising therapeutic target. 

Another HMT with great potential is the disruptor of telomeric silencing like-1 (DOT1L), which plays a major role as a protector of cartilage against OA [67], and in prenatal and postnatal chondrocyte development [68]. DOT1L modulates the histone deacetylase sirtuin 1 (SIRT1) through H3K79 methylation. The presence of this histone mark is reduced in damaged areas of OA cartilage [67]. Additionally, its absence elicits an increased cartilage damage in a DMM-induced OA mouse model [69]. Hence, DOT1L prevents Wnt hyperactivation under non-OA conditions, protecting articular cartilage. In turn, DOT1L levels are regulated by HIF1A, a hypoxia-dependent transcription factor [70] that is able to positively modulate expression of anabolic genes (*COL2A1* and *ACAN*) and negatively regulate catabolic genes (*MMP13* and *NFKB*) [71]. The characteristic hypoxic conditions of healthy cartilage are disrupted in OA, so restoring hypoxic conditions in the joint to promote DOT1L function might be an innovative strategy to combat OA [70].

On the other hand, several studies have demonstrated that HDMTs also contribute to OA pathogenesis. Wang and co-workers elucidated that KDM6A (also known as Ubiquitously transcribed tetratricopeptide repeat protein X-linked or UTX) is required in chondrogenic differentiation of PDLSCs, a promising resource for cartilage regeneration. Depletion of KDM6A resulted in an increased level of transcriptional silencing-related mark H3K27me3 and decreased level of transcriptional activating-related mark H3K4me3 in the *SOX9* promoter. Additionally, KDM6A demethylated H3K27me3 from gene promoter regions of chondrogenic markers (*SOX9*, *COL2A1,* and *ACAN*), allowing its transcription and promoting chondrogenesis [66]. 

Conversely, lysine demethylase 6B or KDM6B (also known as Jumonji domain-containing protein D3 or JMJD3) exhibits high levels in OA and induces the expression of proinflammatory cytokines (IL-6 and TNF-α) and catabolic factors (MMP-13 and ADAMTS-5) by H3K27me3/2 demethylation [72]. These cartilage-degrading effects can be pharmacologically reverted by inhibiting JMJD3 activity, which also promotes the synthesis of crucial components of the ECM, such as COL II and ACAN. In addition, inhibition of JMJD3 via IA injection in DMM mice ameliorates the progression of OA [72]. Jin and collaborators found that JMJD3 is a mechanical stress-responsive protein, since it is upregulated in chondrocytes treated with fluid shear stress (FSS) and OA mouse joint tissues. Moreover, JMJD3 inhibition alleviates FSS-induced damage through H3K27me3 demethylation in the nuclear receptor subfamily 4 group A member 1 (*NR4A1*) promoter, which is related to chondrocyte apoptosis, inhibition of COL2A1 and SOX-9, and induction of COX-2 and MMP-13 expression [73]. Therefore, JMJD3 could be crucial in OA pathogenesis, and it is likely to become a novel therapeutic target in OA. Furthermore, aberrant mechanical forces to which the cartilage might be subjected in OA, such as FSS, can lead to an upregulation of lysine methyltransferase 2B (Kmt2b), which actively trimethylates H3K4. The zinc finger and BTB domain containing 20 (*Zbtb20*) gene shows remarkable H3K4me3 levels, which translates into high levels of Zbtb20 protein. The latter targets the Wnt signalling pathway and promotes OA by promoting cell apoptosis and upregulating inflammatory factors expression (Tnf-α, Il-1β, and Il-6), hypertrophy and catabolism-related markers (Col10, Cox-2, Adamts4, and Adamts5), and cartilage-degrading enzymes (Mmp-3, Mmp-9 and Mmp-13) [74].

### 5.2. Histone Acetylation and Deacetylation in OA

Histone acetylation occurs on K residues located within the N-terminal tails of histones, altering their positive charge. This weakens histone–histone and histone–DNA interactions, resulting in the unwinding of the nucleosomal assembly [75]. Histone acetylation is catalysed by histone acetyltransferases (HATs), whereas the reverse reaction is carried out by histone deacetylases (HDACs). The former is usually associated with activation of gene transcription, while the latter is usually considered a repressive mark [76].

Even though the role of HATs has been poorly characterised in OA, there are a few examples that could potentially be therapeutic targets (Figure 4). First, p300/CBP-associated factor (PCAF) is a HAT that, along with H3K9ac, is upregulated in OA cartilage and in TNF-α-stimulated chondrocytes [77]. PCAF might be an important regulator of the inflammatory response in cartilage due to its implication in the NF-κB pathway [78]. Pharmacological blockade of PCAF activity with salidroside, a natural compound found in the root of *Rhodiola rosea*, supresses inflammation and endoplasmic reticulum stress, which are typical features of OA [77]. Second, bromodomain containing 4 (BRD4) recognises H3K27ac sites to regulate gene transcription. Among these genes is triggering receptor expressed on myeloid cells 1 (*Trem1*), which is upregulated in inflamed areas of the OA synovial membrane [79]. Interestingly, the *Trem1* promoter shows more BRD4 and H3K27ac co-occupancy under mechanical stress, inducing the expression of proinflammatory factors. On the contrary, inhibition of BRD4 counteracts this process, reducing the inflammatory response, and protects TMJ OA rats from cartilage thinning and subchondral bone resorption [80].

Human HDACs can be classified into four classes (Figure 4). Class I, Class II, and Class IV are Zn^2+^-dependent, while Class III is NAD^+^-dependent. Class I consists of HDAC1, -2, -3, and -8. Class II comprises HDAC4, -5, -6, -7, -9, and -10. Class II is further subdivided into Class IIA (HDAC4, -5, -7, and -9) and Class IIB (HDAC6 and -10). On the other hand, Class III includes sirtuins (Sirt1, -2, -3, -4, -5, -6, and -7) [81]. Within the latter category, only Sirt1-3 has potent deacetylase activity, whereas in the case of Sirt4-7, the deacetylase activity is very weak in vitro [82]. Finally, Class IV includes HDAC11.

#### 5.2.1. HDACs with Zn^2+^-Dependent Activity

HDAC1 and HDAC2 share close to 80% amino acid homology, with the greatest divergence in the carboxy-terminal domains (CTDs). The CTDs are not required for catalytic activity, but are necessary for the specific repression of the target genes of both HDACs [83]. Since HDACs cannot bind directly to DNA or histones, it is believed that they associate with other transcription factors via CTDs. Thus, these transcription factors recruit HDACs to the promoters of their target genes [75]. A study revealed that HDAC1 and HDAC2 are both upregulated in OA cartilage and repress cartilage-specific genes in human chondrocytes [83]. They do this by binding to the transcription factor Snail via CTDs and each has specific targets. For example, the CTD of HDAC1 is essential for the repression of *COL2A1* and *COL9A1*, while the CTD of HDAC2 allows the repression of *COL2A1* and cartilage oligomeric protein (*COMP*). However, it is likely that there are domains other than CTDs in each HDAC responsible for *ACAN*, *COL11A1,* and dermatopontin (*DPT*) repression [83].

HDAC4 is expressed in prehypertrophic chondrocytes and functions by supressing the activity of Runx2, which is the master regulator of osteogenesis and is required for chondrocyte hypertrophy. This demonstrates that HDAC4 is a major player not only in skeletogenesis, where it prevents premature endochondral bone formation, but also in chondrocyte hypertrophy, through Runx2 inhibition [84]. 

Fu and collaborators observed that mechanical loading activates HDAC6 to inhibit tubulin acetylation and IL-1β-induced polymerisation. Hence, IL-1β-induced cilia elongation in primary articular chondrocytes is interrupted. Accordingly, inhibition of HDAC6 restores cilia elongation and promotes the inflammatory effect of IL-1β. These data suggest that the anti-inflammatory effects of mechanical loading are mediated by HDAC6-catalysed deacetylation of tubulin [85]. Despite this, HDAC6 is proven to be upregulated on the articular surface of DMM mice and its overexpression results in mitochondrial dysfunction accompanied by elevated ROS production and glycosaminoglycan (GAG) loss. Inhibition of HDAC6 with Tubastatin A counteracted ROS increase and promoted GAG production in the joint [86].

#### 5.2.2. HDACs with NAD^+^-Dependent Activity: The Sirtuins

Emerging evidence has positioned sirtuins as other promising therapeutic targets (Figure 4). SIRT1 protein is excised by cathepsin B in TNF-α-stimulated human OA chondrocytes, yielding a N-terminal fragment (NT) and a C-terminal fragment (CT). As a consequence, its deacetylase activity is lost [87]. Batshon et al. observed an increase in NT/CT SIRT1 ratio in moderately severe OA in mouse and human cartilage samples. This ratio was elevated in humans with early-stage OA. Additionally, when they administered UBX0101 (Unity Biotechnology, San Francisco, CA, USA), the NT/CT SIRT1 ratio was reduced. Thus, the NT/CT SIRT1 ratio in serum might serve as an OA biomarker to measure the disease severity and the efficacy of senolytic drugs [88]. SIRT1 has also been related to autophagy. Sacitharan and collaborators observed a SIRT1 reduction in cartilage from OA and healthy aged patients. Silencing of SIRT1 induced the downregulated expression of chondrogenic markers, whereas SIRT1 activation led to increased autophagy by the deacetylation of critical autophagy markers, such as Beclin1, autophagy related 5 (ATG5), autophagy related 7 (ATG7), and microtubule-associated protein 1A/1B-light chain 3 (LC3). In fact, SIRT1 binds to these proteins, which suggests that SIRT1 directly interacts and activates autophagy in chondrocytes [89]. These data correlate with Lu et al.’s findings, who explored the role of the SIRT1/phosphatase and tension homolog deleted from chromosome 10 (PTEN)/epidermal growth factor receptor (EGFR) axis in OA. Their results showed that SIRT1 restoration represses the ubiquitination of EGFR by downregulating PTEN and suppresses ECM degradation, exerting a chondroprotective role [90].

SIRT2 has been associated with diabetic OA. In fact, SIRT2 levels are lowered in patients with diabetic OA, whereas acetylation of H3K9, H3K14, and H3K56 is increased. The same results were obtained in an in vitro model of diabetic OA, in which SIRT2 expression progressively decreased as glucose concentration increased. This translated into the accumulation of H3ac, along with an increase in ROS production and *MMP13*, *ADAMTS4,* and *ADATMS5* expression. On the contrary, upregulation of SIRT2 protects against OA by reducing H3ac, oxidative stress, and inflammation [91]. Another example is SIRT3, which is shown to exert protective effects against OA via inhibiting a key signalling pathway of autophagy: the PI3K/Akt/mTOR pathway. SIRT3 overexpression prevents the expression of catabolic molecules, such as MMP-3, MMP-13, COX2, and iNOS, while it enhances the expression of anabolic molecules, including COL2A1, SOX-9, and ACAN. Strikingly, SIRT3 protects IL-1β-treated chondrocytes from apoptosis and regulates mitochondrial membrane potential, thereby protecting chondrocytes of mitochondrial dysfunction [92]. Conversely, another recent study supports the notion that the function of Sirt3 in mice might be context-dependent. The authors found that depleting *Sirt3* in mice cartilage, which were previously fed with a high fat diet (HFD), induced a chondroprotective effect and prevented synovitis. *Sirt3* expression leads to the decreased production of glycolytic proteins. Moreover, it enhances mitochondrial respiration and fatty acid metabolism, thereby promoting HFD-induced OA. However, the deletion of *Sirt3* plays a detrimental role in chondrogenesis when using a murine bone marrow stem cell pellet model, so its role on cartilage homeostasis has yet to be fully elucidated [93].

Although not many studies have been conducted, SIRT4 has been associated with cartilage protection against OA. Treatment of chondrocytes with SIRT4 promotes COL2A1 and ACAN secretion and the expression of the antioxidant enzyme superoxide dismutase type 1 (SOD1) while suppressing the expression of proinflammatory markers (MMP-13, IL-6, and TNF-α) and ROS production [94]. Interestingly, Sirt5 was initially considered as a lysine deacetylase, but experimental data shows that its deacetylase activity is considerably lower compared to its demalonylase and desuccinylase activity [82]. In fact, Sirt5 catalyses lysine malonylation (MaK), a PTM which is upregulated in obese *db*/*db* mice (a type 2 diabetes mouse model). Moreover, DMM mice also showed an age-dependent increase in MaK levels. The authors next proved that Sirt5 deficiency caused a global hypermalonylation of proteins and reduced glycolysis and mitochondrial respiration rate, suggesting that Sirt5 is involved in the regulation of chondrocyte metabolism and might contribute to OA [95].

Previous studies defend that SIRT6 confers protection against aging-related pathologies and promotes longevity [96]. Recently, Collins and collaborators linked SIRT6 to the OA context, demonstrating that it acts as an important regulator of chondrocyte redox balance [97]. SIRT6 displays an age-dependent deacetylase activity, as H3K9ac basal levels are increased in human chondrocytes from older adults. Likewise, oxidative stress also contributes to SIRT6 activity decline. SIRT6 overexpression in chondrocytes increased the levels of the antioxidant proteins peroxiredoxin 1 (PRX1) and sulfiredoxin (SRX). Parallelly, it reduced the levels of thioredoxin interacting protein (TXNIP), an inhibitor of antioxidant activity. The antioxidant nature of SIRT6 could be useful in OA, where oxidative stress plays a crucial role on its pathogenesis [97]. In agreement with these findings, another study showed reduced levels of SIRT6 in knee synovial tissues of OA patients. Through a DMM murine model, the authors observed that *Sirt6* overexpression attenuated OA progression by decreasing the levels of Tnf-α, Il-1β, and Il-4 in serum [98].

In contrast, Sirt7 might play the opposite role to SIRT6. Korogi et al. discovered that *Sirt7* knockout mice are resistant to the development of OA induced by aging or mechanical loading [99]. Accordingly, *Sirt7* knockdown ATDC5 cells exhibited higher levels of GAG content and upregulated expression of *Col2a1* and *Acan*. On the contrary, the attenuation of *Sirt7* induced an enhanced *Sox9* expression that was reverted after Sirt7 overexpression. However, Sirt7 overexpression was not accompanied by an increased acetylation of Sox-9, which suggests that Sirt7 might be suppressing the transcriptional activity of Sox-9 by succinylation or acylation instead by acetylation. On the other hand, Wu et al. found that SIRT7 is supressed in OA and this deficiency impairs autophagy by decreasing the expression of COL II, Unc-51 like autophagy activating kinase 1 (ULK1), LC3, and Beclin1 [100]. Nevertheless, Sirt7 could be a useful therapeutic target against OA.

## 6. Non-Coding RNA (ncRNA) Modifications in OA

### 6.1. MicroRNAs (miRNAs)

MicroRNAs (miRNAs) are small single-stranded ncRNAs of 21–25 nucleotides in length that mostly function as post-translational repressors [101], intervening in the control of biological processes such as essential as cell proliferation, apoptosis, differentiation, or organogenesis [102]. The field of miRNAs has experienced considerable growth in recent years, since the characteristics of these small molecules hold great promise for biomedical and clinical research. In this section, some of the most relevant miRNAs in OA will be reviewed. In addition, other examples are included in Table 1.

Among them, miR-140 is specifically expressed in cartilage [103,104] and exerts a critical role in chondrogenesis [105], the reason why it has attracted increasing interest in the field of OA. MiR-140 is located in an intronic region of the WW domain containing the E3 ubiquitin protein ligase 2 (*WWP2*) gene, which can generate two variants: miR-140-5p and miR-140-3p. These, although derived from the same precursor transcript, have different target genes and both are downregulated in OA chondrocytes [106]. The regulatory sequence is located upstream of the miR-140 precursor, and it comprises binding sites for different transcription factors, such as mothers against decapentaplegic homolog 3 (SMAD3) [106]. Eight highly methylated CpG sites were found in OA chondrocytes within this regulatory region. Hypermethylation of these CpG sites downregulates miR-140-5p expression through changes in SMAD3 binding affinity to *MIR140* regulatory sequence, thereby influencing the expression of specific target genes (*MMP13* and *ADAMTS5*) [107]. In OA cartilage, miR-140 expression is markedly decreased compared to normal cartilage, which promotes the expression of genes with detrimental roles in OA [108,109]. Exposure of chondrocytes to IL-1β significantly lowers miR-140 expression and, in parallel, causes an elevated expression of the catabolic genes *MMP13* and *ADAMTS5* [108]. In line with this, overexpression of miR-140 in human OA chondrocytes promotes COL2A1 expression and supresses MMP-13 and ADAMTS-5 expression at the protein level [110]. Another study showed that mice lacking miR-140 (*Mir140*^-/-^) developed an age-related OA-like pathology and transgenic mice overexpressing miR-140 exhibited resistance to antigen-induced arthritis, reinforcing the role of this miRNA as a cartilage homeostasis regulator. In addition, the authors revealed that miR-140 is a direct repressor of *Adamts5*. *Adamts5* expression was notably increased in *Mir140*^-/-^ mice chondrocytes and cartilage explants exhibited significant PG loss due to the absence of miR-140 [111].

MiR-146a is also a highlighted miRNA, as it is strongly expressed in cartilage with low Mankin score, especially in chondrocytes located in the superficial layer [112]. The promoter of *MIR146A* presents three hypermethylated CpG sites in OA synoviocytes that impair NF-κB binding to this region. This silences miR-146a expression which, in turn, induces the expression of inflammatory factors, such as *IL1B*, *IL6,* or *TNFA* among others [107]. Moreover, miR-146a expression can also be induced in vitro by IL-1β-stimulation in normal articular chondrocytes [112]. Exposure of OA chondrocytes to cyclic low HP significantly downregulates the expression of miR-146a, miR-34a, miR-181a, *MMP13*, *ADAMTS5*, superoxide anion, and various antioxidant enzymes, while it upregulates *COL2A1* levels [113]. On the contrary, the application of continuous static pressure induces the opposite effect and promotes apoptosis. The same study evaluated the silencing of miR-146a, along with the other miRNA mentioned above, finding that inhibiting the expression of these miRNA enhances the protective effect exerted by low HP and counteracts the detrimental effects of continuous static pressure. Collectively, these results imply that miRNAs could act as mediators of the molecular mechanisms induced by HP on chondrocyte metabolism [113]. The other strand of miR-146, miR-146b, has also been found to be upregulated in human OA chondrocytes [114]. Apart from this, miR-146b was progressively downregulated during the chondrogenic differentiation of human bone marrow-derived skeletal stem cells (SSCs). Moreover, increased miR-146b levels induced a decrease in SOX-5 levels, an important transcription factor during early chondrogenesis. Taken together, these data highlight the therapeutic potential that this miRNA could have if used as a chondrogenic enhancer to prevent the early onset of OA [114].

Endisha and collaborators detected high levels of miR-34a-5p in knee joint tissues obtained from total knee replacement (TKR) patients and DMM mice. Additionally, they reported systemically increased expression in late-stage OA. Then, they explored the role of miR-34a-5p in OA pathogenesis and found that it exerts negative effects in the joint, promoting the expression of inflammatory, profibrotic, and autophagy markers. Moreover, IA injection of miR-34-5p mimic enhanced PG loss, apoptosis, and synovial ECM deposition, thereby creating an OA-like phenotype [115]. Similarly, miR-181a-5p expression is increased in human and mouse knee OA cartilage and promotes cartilage breakdown. Treatment of OA joints in rat and mice with locked nucleic acid miR-181a-5p antisense oligonucleotides (LNA-miR-181a-5p ASO), which are modified oligonucleotides that specifically target miR-181a-5p, attenuates the expression of catabolic, hypertrophic, and cell death-related molecules, having a protective effect on cartilage [116]. Thus, targeting miR-34a-5p and miR-181-5p could be a powerful strategy to attenuate OA progression.

Several miRNAs have been identified in an OA context. However, the functions exerted by each of them have not yet been fully described. An example that may illustrate this is miR-204. Opposite roles have been reported for this miRNA. While Kang et al. found that miR-204 is upregulated in OA cartilage and it hampers PG synthesis and promotes chondrocyte senescence under stress conditions [117], Huang et al. defend that miR-204, along with its homolog miR-211, exerts beneficial effects on cartilage homeostasis and supresses OA [118]. In fact, miR-204/-211 deficiency in mesenchymal progenitor cells (MPCs) increased Runx2 and OA catabolic markers expression, promoting OA in mice joint tissues [118].

MiRNAs can also interact with histones and enzymes that catalyse PTMs, creating complex epigenetic regulatory mechanisms. For instance, miR-140 targets the 3’ UTR region of the HDAC4 mRNA, inhibiting its transcription [104]. As mentioned in previous sections, HDAC4 interacts with *Runx2* and inhibits its activity, preventing chondrocytes from hypertrophy and promoting chondrocyte differentiation. This means that miR-140 is an indirect regulator of *Runx2* via HDAC4 silencing [84]. Likewise, miR-92a-3p targets the 3’ UTR region of HDAC2 mRNA and downregulates HDAC2. The expression of cartilage-specific components is favoured by miR-92a-3p, as it enhances H3 acetylation in *COL2A1*, *ACAN*, *COMP,* and *DPT* promoters. In fact, OA cartilage exhibits high levels of HDAC2 and low levels of miR-92a-3p, a scenario that promotes cartilage degeneration [119]. On the other hand, EZH2 increases H3K27me3 at the *MIR138* promoter, inhibiting miR-138 expression and accelerating cartilage destruction by upregulating syndecan 1 (SDC1) expression in OA [120]. MiR-138 was previously reported to be greatly reduced in OA cartilage, which promotes cartilage damage by targeting p65, a NF-κB subunit [121].

MiRNAs have great potential as biomarkers of OA, as has been proven by several authors. Yin et al. examined the expression of miR-140-5p and miR-140-3p in the SF from OA knees, finding that both miRNAs were significantly downregulated compared with levels exhibited by SF from non-OA knees. In fact, the expression of miR-140-5p and miR-140-3p was negatively correlated with severity of OA [122]. Additionally, Si and collaborators showed that IA injection of miR-140 regulates ECM homeostasis in an OA rat model, attenuating the progression of the disease [110]. Inversely, miR-210 is found in large amounts in the SF from early- and late-stage OA patients and it might be useful in the early diagnosis of OA [123]. However, obtaining SF is still an invasive procedure, so other options have also been investigated. Okuhara et al. aimed to investigate the expression patterns of various miRNAs in peripheral mononuclear blood cells from OA patients. They found that miR-146a and miR-223 are intensely expressed in early OA, whereas miR-155 is associated with late OA and its expression increases as the Kellgren–Lawrence (KL) grade does. Moreover, miR-146a expression exhibited a positive correlation with age [124]. In serum samples from women with mild to moderate knee OA, the variant miR-146a-5p is increased in comparison with healthy controls. On the other hand, miR-186-5p was found to be indicative of knee OA incidence, since women with higher miR-186-5p expression were more likely to develop knee OA over the next 4 years [125]. Giordano and collaborators showed that circulating miRNAs in serum, particularly miR-146a-5p, miR-145-5p, and miR-130-3p, could serve as predictive biomarkers of pain after TKR surgery, as they were incremented in patients who suffered low pain relief compared to those who showed high pain relief after surgery [126].

Mounting evidence indicates that miRNAs could be a potential therapeutic target against OA. In combination with other strategies, such as the use of stem cells, it could succeed in halting and repairing joint deterioration or even slowing the disease in its early stages [127]. For this reason, research in this field is an essential tool for designing new treatments.
pharmaceuticals-16-00156-t001_Table 1Table 1Overview of the most relevant miRNAs in OA.miRNATarget GenesExpression in OAEffective RoleSpeciesReferencesmiR-27a*PI3K*↑Promotes apoptosis and autophagy by ↓ PI3KHuman[128]miR-34a-5p*Pparg, Cadm1, Abcc5, Reck,**Maoa, Adgrg2*↑↑ Inflammatory, profibrotic, and autophagy markersHuman/Mouse[115]IA injection → OA-like phenotypemiR-92a-3p*HDAC2*↓↑ H3ac in *ACAN*, *COMP*, *COL2A1* promotersHuman[119]miR-93*Tlr4*↓Inhibits lipopolysaccharide (LPS)-induced inflammation and cell apoptosis by supressing the Tlr4/NF-κB pathwayMouse[129]miR-103*Sphk1*↑Promotes apoptosis by ↓ PI3K/Akt pathwayHuman/Rat[130]miR-132-↓↑ Cell proliferation, ↓ apoptosis through the PTEN/PI3K/AKT signalling pathwayHuman/Rat[131]miR-132-3p*Adamts-5*↓↑ *Sox9*, *Acan*, *Col2a1* levels during chondrogenic differentiation of mesenchymal stem cellsRat[132]Promotes PG deposition*Pten*↓↑ Cell proliferation, ↓ apoptosis and inflammation through the Pten/PI3K/Akt signalling pathway in TMJ OA chondrocytesRat[133]miR-138*SDC1*↓Avoids cartilage destructionHuman/Mouse[120]p65Promotes cartilage protection by ↓ p65Human[121]miR-140*ADAMTS5, ACAN*↓Implicated in chondrogenesis, promotes ECM maintenanceHuman[108]Absence → PG loss, fibrillation of cartilageMouse[111]miR-146a
↑Potential mediator of chondrocyte metabolism in response to HPHuman[113]-↑ In early OAHuman[124]miR-146b*SOX5*↑↓ During the chondrogenic differentiation of bone marrow-derived SSCsHuman[114]miR-155-↑↑ In late-stage OAHuman[124]miR-181a-5p-↑↑ In early OAHuman[124]miR-186-5p-↑Indicative of knee OA incidenceHuman[125]miR-193b-5p*HDAC7*↓↓ *MMP3*, *MMP13*Human[134]miR-193b-3p*HDAC3*↓↑ H3ac in *ACAN*, *SOX9*, *COL2A1*, *COMP* promotersHuman[135]miR-204*SLC35D1*, *CHSY1*, *CHST11*, *HAPLN1*↑↓ PG synthesis and promotes chondrocyte senescenceHuman/Mouse[117]-miR-204/-211 deficiency in MPCs ↑ Runx2 and OA catabolic markersMouse[118]miR-210-↑↑ In SF from early- and late-stage OA patientsHuman[123]miR-211*Runx2, Bmpr2, Fas, Pten, Reck, Jag1, Tgfbr2*↓↓ Mineralisation and OA subchondral bone osteoblast gene expressionRat[136]-miR-204/-211 deficiency in MPCs ↑ Runx2 and OA catabolic markersMouse[118]miR-214-3p*IKBKB*↓Inhibits the activation of NF-κB pathwayHuman/Mouse[137]IA injection → Attenuation of OAmiR-218-5p*PIK3C2A*↑↓ Viability, ↑ apoptosis, inflammation (↑ *IL6*, *TNFA*, *COX2*), ECM catabolism (↑ *MMP13*, *ADAMTS5*, ↓ *COL2A1*) through the PI3K/AKT/mTOR pathwayHuman[138]*HRAS*↑↓ Cell proliferation, ↑ inflammation and ECM degradationHuman[139]miR-223-↑↑ In early OAHuman[124]miR-335-5p-↓↑ Viability, GAG content, autophagy markersHuman[140]↓ Inflammatory markers, apoptosismiR-455-5p*EPAS1*↓Suppression of the catabolic factor HIF-2α (encoded by *EPAS1*)Human/Mouse[141]miR-455-3p↓IA injection → ↓ Cartilage destructionmiR-485-3p*Notch2*↓↓ ECM degradation, inflammation, oxidative stress, apoptosisHuman[142]Inhibits NF-κB pathway activation by targeting Notch2


### 6.2. Circular RNAs (circRNAs)

Circular RNAs (circRNAs) are another type of ncRNAs which are ubiquitous, stable and conserved among mammals. CircRNAs derive from the alternative splicing of pre-mRNA, which results in them having their 3′ and 5′ ends covalently linked [143]. Some circRNAs can behave as competing endogenous RNAs (ceRNAs) that naturally sequester miRNAs, acting as a ‘sponge’ and inhibiting miRNA activity [144]. Although the function of circRNAs has not been fully elucidated, this characteristic feature makes them potential suppressors of those miRNAs that are implicated in the development of diseases. In the present section, some of the most relevant circRNAs in OA will be discussed, which are included in Table 2.

Liu and collaborators identified 71 differentially expressed circRNAs between OA and healthy knee cartilage samples. In OA cartilage, there were 16 upregulated and 55 downregulated circRNAs. Among these, circRNA_100876 (circRNA-CER), circRNA_100086, circRNA_101178, and circRNA_101914 were overexpressed in OA. The authors focused on circRNA-CER (chondrocyte extracellular matrix-related circRNA) and found an increased expression after IL-1 and TNF-α stimulation of chondrocytes. Parallelly, *MMP13* expression was also increased. Further investigation revealed that circRNA-CER possesses binding sites for miR-636, miR-665, miR-217, miR-646, and miR-136. Strikingly, miR-136 also binds to the 3’ UTR of *MMP13*. Therefore, circRNA-CER is targeted by *MMP13*-targeting miRNAs [145]. Furthermore, circ_0136474 was found to be present at high levels in knee OA cartilage. This overexpression leads to the downregulation of miR-127-5p expression, which directly targets the 3’ UTR region of *MMP13*. In fact, silencing of miR-127-5p positively regulates the expression of MMP-13 both at the mRNA and protein levels, while miR-127-5p overexpression leads to an MMP-13 decrease. The higher level of MMP-13 caused by the overexpression of circ_0136474 promotes cell apoptosis and suppresses cell proliferation by inhibiting miR-127-5p [146]. Additionally, Zhou et al. screened the circRNA profile in IL-1β-treated mouse articular chondrocytes and found 255 differentially expressed circRNAs. Of these, 119 were upregulated and 136 were downregulated [147]. After that, the same group analysed one of the upregulated circRNAs, called circ_15898 or circRNA_Atp9b. They found that circRNA_Atp9b modulates Col II, Mmp-13, Cox-2, and Il-6 expression levels by directly targeting and sponging miR-138-5p. Thus, low levels of circRNA_Atp9b might have a protective effect against inflammation and ECM destruction [148]. 

Another study showed the downregulation of circ_00008667, circ_0072568, circ_0008365, circ_0020093, circ_0110251, and circ_0001103 in IL-1β and TNF-α-stimulated human chondrocytes. Among these, knockdown of circ_0008365 also denoted circSERPINE2 and promoted apoptosis and ECM catabolism by increasing MMP-3, MMP-13, and ADAMTS4 expression and decreasing SOX-9, COL II, and ACAN expression. Moreover, circSERPINE2 acts as a sponge for miR-1271, which in turn targets E26 transformation-specific (ETS)-related gene (ERG), a molecule implicated in the pathogenesis of OA. An OA rabbit model confirmed the protective effects of circSERPINE2, since its IA injection contributed to ECM anabolism and alleviated OA [149]. 

There is further evidence that circRNAs could have great potential as biomarkers or therapeutic targets in OA. Another study compared the expression profile of circRNAs in SF from OA and healthy patients by microarray analysis. The results showed 47 differentially expressed circRNAs, 29 of which were upregulated and 18 were downregulated. Circ_0104873, circ_0104595, and circ_0101251 were significantly and positively correlated with radiographic and symptomatic OA severity [150]. Xiang et al. also highlighted the implications of circRNAs in OA synovitis. They identified 122 differentially expressed circRNAs through RNA sequencing (RNA-seq), which included 89 downregulated and 33 upregulated circRNAs. The results confirmed that circ_0001979, circ_0005406, circ_0008172, circ_0015260, and circ_0077425 were downregulated, while circ_0037658 was upregulated in OA synovial tissue [151]. On the other hand, Zhou and collaborators assessed the expression of the circRNA ciRS-7 and miR-7 in the plasma of OA and healthy patients, finding a lower expression of ciRS-7 and higher levels of miR-7 in OA subjects. An OA cell model revealed that the ciRS-7/miR-7 axis modulates apoptosis and inflammation in IL-1β-treated chondrocytes. In fact, apoptosis and inflammation were promoted by ciRS-7 suppression and miR-7 upregulation, which suggests that the ciRS-7/miR-7 might be implicated in OA development [152].
pharmaceuticals-16-00156-t002_Table 2Table 2Overview of the most relevant circRNAs in OA.circRNAExpression in OATarget miRNAsEffective RoleSpeciesReferencescircRNA_Atp9b↑miR-138-5p↓ Col II Mouse[147,148]↑ Mmp-13, Cox-2, Il-6circRNA-CER↑miR-636miR-665miR-217miR-646miR-136↑ *MMP13* expression by binding miR-136Human[145]circSERPINE2↓miR-1271↑ MMP-3, MMP-13, ADAMTS4 Human/Rabbit[149]↓ SOX-9, COL II, ACANIA injection → Attenuation of OA, ↑ ECM anabolismciRS-7↓miR-7↑Apoptosis and inflammationHuman[152]circ_0045714↓miR-218-5p↓IL-6, IL-8, MMP-13, ADAMTS-4↑COL II, ACANHuman[139]circ_0136474↑miR-127-5p↑ MMP-13, apoptosisHuman[146]↓ Cell proliferationcirc_0104873↑
Positively correlated with radiographic and symptomatic OA severity

circ_0104595↑-Human[150]circ_0101251↑





### 6.3. Long Non-Coding RNAs (lncRNAs)

Long non-coding RNAs (lncRNAs) are transcripts of more than 200 nucleotides in length that apparently do not encode proteins. They are involved in various functions and their overexpression, deficiency, or mutation can lead to certain diseases [153]. Some examples of lncRNAs involved in OA are described below and are listed in Table 3.

An RNA-seq analysis in samples from human hip cartilage identified 1692 Ensembl lncRNAs located within intergenic regions (long intergenic non-coding RNAs, lincRNAs), of which 198 were differentially expressed between OA and healthy patients. PART1, LINC01139, and NORAD were among the most upregulated lincRNAs in OA, whereas LUCAT1, MEG3, and LINC01679 were included in the most downregulated. A similar approach was followed with knee OA samples. CRNDE, MIR22HG, and LINC01614 were the most upregulated lincRNAs in the damaged area, while MEG3, ILF-AS1, and LINC01089 were the most downregulated [154]. Hoolwerff et al. also conducted a study using RNA-seq to identify differentially expressed lncRNAs in knee and hip cartilage samples. They found a total of 5053 lncRNAs, of which 191 were differentially expressed between damaged and preserved OA cartilage, with intergenic and anti-sense lncRNAs being prone to regulate mRNAs in *cis* in OA cartilage [155].

HOX transcript antisense intergenic RNA (HOTAIR) is a widely studied lncRNA that is dysregulated in several types of cancer, being deeply implicated in cancer pathogenesis, progression and drug resistance [156]. Although not as extensively studied in OA, HOTAIR appears to be implicated in this pathology as well. HOTAIR was significantly present at higher levels in OA knee cartilage samples [157]. A chondrocyte in vitro model revealed that HOTAIR promotes apoptosis and restricts autophagy by targeting and sponging miR-130-3p. HOTAIR silencing exerted the opposite effect by inducing the expression of pro-apoptotic protein BAX and cleaved-Caspase-3 and decreasing the expression of the anti-apoptotic protein B-cell lymphoma-2 (BCL-2). Consistent with this, HOTAIR silencing increased the ratios of LC3 II to LC3 I and reduced the levels of p62, which suggests that HOTAIR is also implicated in the regulation of autophagy [157]. Another study conducted an OA mouse model and an IL-1β-induced chondrocyte model and discovered that the upregulation of HOTAIR is associated with the pathogenesis of OA. The expression of HOTAIR and miR-20b exhibited a negative correlation, HOTAIR being upregulated and miR-20b downregulated. Additionally, HOTAIR silencing reduced Pten, apoptosis, and the expression of ECM catabolic molecules. Further investigation in IL-1β-induced chondrocytes revealed that Pten is a target of miR-20b and HOTAIR participates in the miR-20b/Pten axis via sponging miR-20b, which has a detrimental effect in OA [158]. On the other hand, Jiang and collaborators compared the lncRNAs p50-associated COX-2 extragenic RNA (PACER) and HOTAIR in plasma from OA and healthy patients and found a clear dysregulation of both. Plasma levels of PACER were lower in OA than in healthy patients, while HOTAIR levels were higher in OA. Further research showed that overexpression of PACER led to a significant downregulation of HOTAIR expression. However, HOTAIR overexpression did not alter PACER expression, but led to increased apoptosis. This suggests that PACER is an upstream inhibitor of HOTAIR in the CHON-001 chondrogenic cell line [159].

In contrast to HOTAIR, metastasis-associated lung adenocarcinoma transcript 1 (MALAT1) is a chondroprotective lncRNA. As Gao et al. demonstrated, MALAT1 blocked the activation of the c-Jun N-terminal kinase (JNK) signalling pathway, which reduced the inflammatory response in rat chondrocytes. IL-1β-treated chondrocytes exhibited low levels of MALAT1, accompanied by low proliferation rates and increased apoptotic events. Meanwhile, MALAT1 overexpression promotes chondrocyte proliferation and viability. Additionally, it elevates the expression of Col II and reduces Mmp-13 expression [160]. However, MALAT1 silencing in LPS-treated chondrocytes from rats produced a pronounced reduction of Il-6, Cox-2, and Mmp-13, while increasing Col II expression. In addition, it induced apoptosis and reduced proliferation and viability. Moreover, MALAT1 depletion significantly increased miR-146a expression, which directly targets phosphoinositide 3-kinase (PI3K). This suggests that MALAT1 exerts a protective role in LPS-treated chondrocytes against apoptosis and inflammation by sponging miR-146a, which in turn targets the PI3K/Akt/mTOR signalling pathway [161].
pharmaceuticals-16-00156-t003_Table 3Table 3Overview of the most relevant lncRNAs in OA.lncRNAExpression in OAEffective RoleSpeciesReferencesCRNDE↑↑ Upregulated in knee OAHuman[154]HOTAIR↑↑ Apoptosis, ↓ autophagy by targeting miR-130-3pHuman[157]Inhibits proliferation of IL-1β-induced chondrocytes ↑ Apoptosis and ECM degradation by targeting miR-20bMouse[158]↑ ApoptosisHuman[159]ILF-AS1 ↓↑ Downregulated in knee OAHuman[154]KCNQ1OT1↓↑ Viability, ↓ Apoptosis, inflammation, and ECM catabolism↑ PIK3C2A and activated the PI3K/AKT/mTOR pathway by targeting miR-218-5pHuman[138]LINC01089↓↑ Downregulated in knee OAHuman[154]LINC01139↑↑ Upregulated in hip OAHuman[154]LINC01614↑↑ Upregulated in knee OAHuman[154]LINC01679↓↑ Downregulated in hip OAHuman[154]LUCAT1↓↑ Downregulated in hip OAHuman[154]MALAT1↓Blocks the activation of JNK signalling pathway↓ Col II, ↑ Mmp-13 in IL-1β-induced chondrocytesRat[160]↑↓ Col II, ↑ Mmp-13, Cox-2, Il-6 by targeting miR-146a in LPS-induced chondrocytesRat[161]MEG3↓↑ Downregulated in hip OAHuman[154]↑ Downregulated in knee OAMIR22HG↑↑ Upregulated in knee OAHuman[154]NORAD↑↑ Upregulated in hip OAHuman[154]PACER↓Inhibits HOTAIR and apoptosis in CHON-001Human[159]PART1↑↑ Upregulated in hip OAHuman[154]


## 7. Conclusions

Considering recent evidence, epigenetics is deeply involved in the development of OA at molecular level and could serve as a powerful tool to attenuate the progression of the disease (Figure 5). However, further research is still needed to fully understand the mechanisms of epigenetic regulation of OA and to be able to use them in clinical practice. In particular, the integrative study of all the epigenetic mechanisms that have been reviewed is key to gain a better perspective on the implications of epigenetics in the development of OA, e.g., the interaction of non-coding RNAs with histones and their PTMs, chromatin remodelling, or how histone PTMs influence interactions between nucleosomes. These are still unknown aspects of OA, but they could be important in its pathogenesis and useful in the search for novel treatments or biomarkers.

## Figures and Tables

**Figure 1 pharmaceuticals-16-00156-f001:**
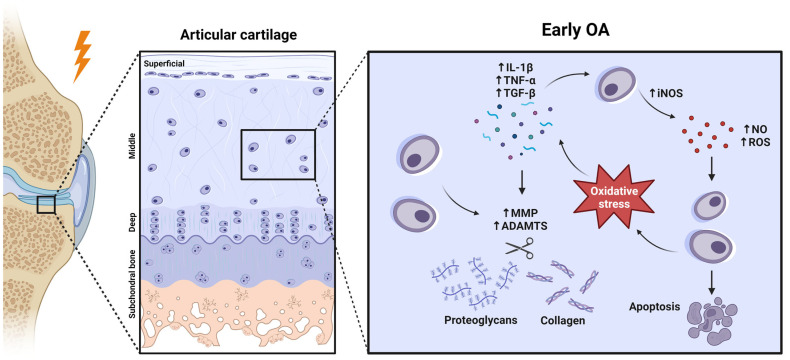
Early OA development at a molecular perspective. Created with BioRender.com, accessed on 14 January 2023.

**Figure 2 pharmaceuticals-16-00156-f002:**
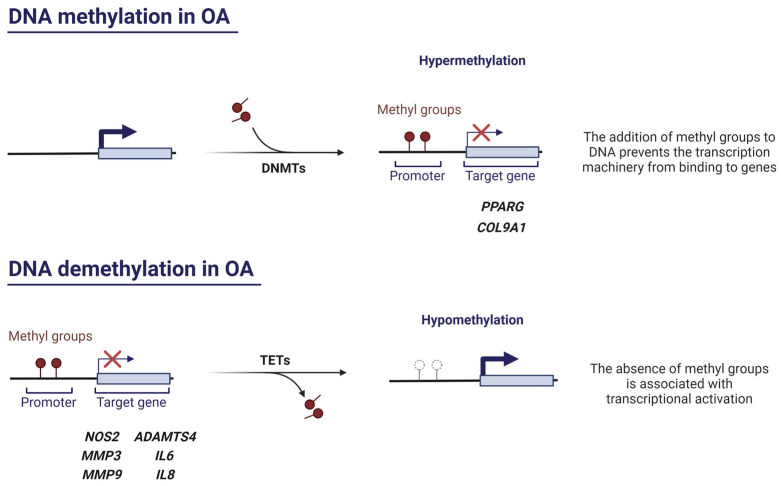
The role of DNA methylation and demethylation in OA. Created with BioRender.com, accessed on 13 January 2023.

**Figure 3 pharmaceuticals-16-00156-f003:**
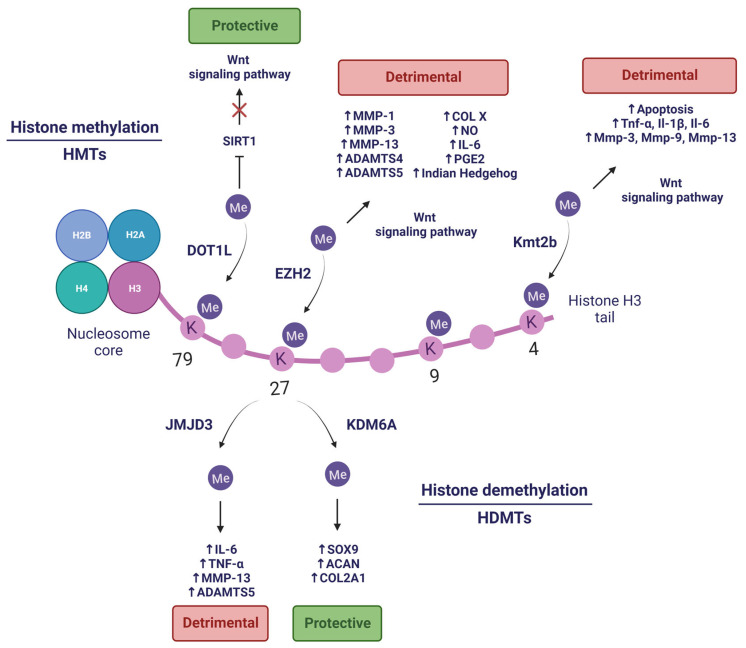
Histone methylation and demethylation in OA. Created with BioRender.com, accessed on 14 January 2023.

**Figure 4 pharmaceuticals-16-00156-f004:**
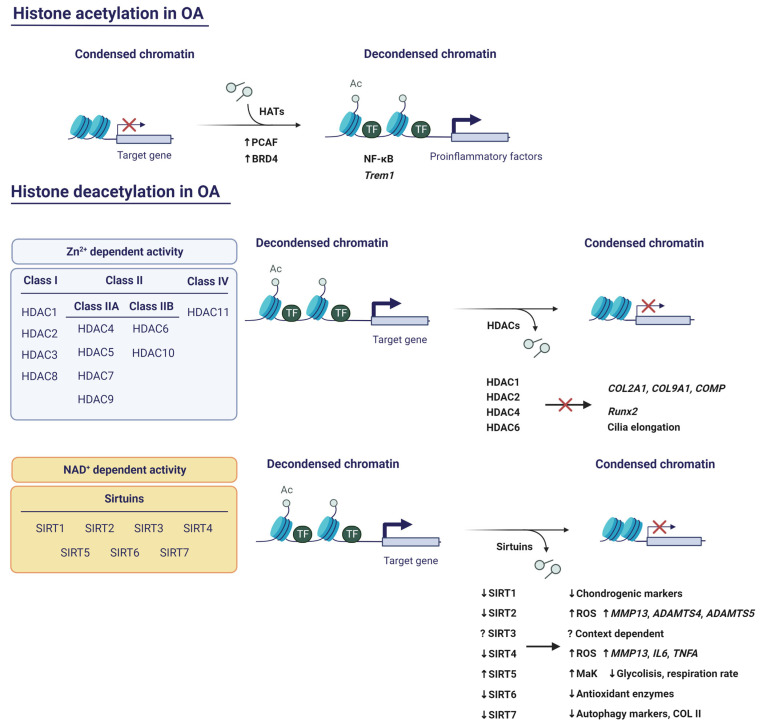
Implications of histone acetylation and deacetylation in OA. Created with BioRender.com, accessed on 13 January 2023.

**Figure 5 pharmaceuticals-16-00156-f005:**
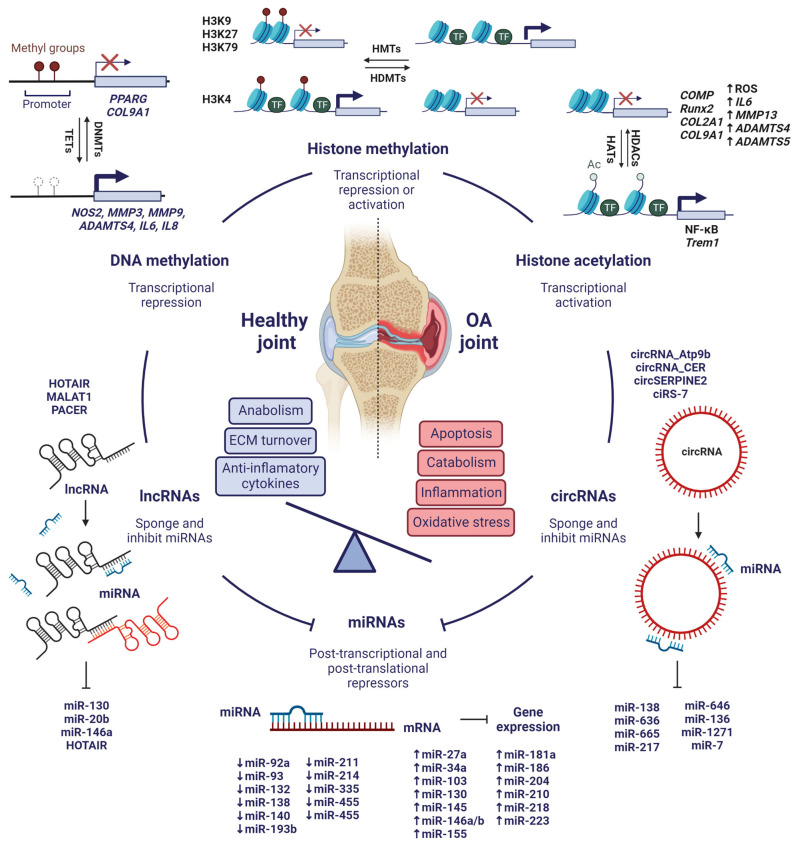
Integrative view of the most significant epigenetic mechanisms involved in the pathogenesis of OA. Created with BioRender.com, accessed on 15 January 2023.

## Data Availability

Data is contained within the article.

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
