# Peer review of "Epigenetics as a Therapeutic Target in Osteoarthritis"

_pharmaceuticals, 2023, doi:10.3390/ph16020156_

Round 1
Reviewer 1 Report
The authors reviewed the importance of epigenetics in osteoarthritis. Focused on three classical epigenetic regulatory mechanisms, including DNA methylation, histone modifications, and changes in non-coding RNAs. Overall, the structure of the manuscript is good. I have several comments which may help to improve the interpretations.
1. Institution address doesn't seem to be in English.
2. Line 40 ‘. may be’. The period should be changed to a comma.
3. I don’t understand the meaning of ‘rs75621460 may also represent a powerful therapeutic target’ in line 213. The SNP could be the target, or you mean the gene TGFB1 ?
4. Line 268-272, ref 60 shows dexamethasone decreases MMP13, COL2A1 and enhances ACAN, while ref 61 shows dexamethasone decreases Col2a1, Acan and enhances Mmp13. Need a better explanation here.
5. Line 492, ‘MIR140’; Line 496 and 498, ‘miR-140’; Line 540 ‘Mir140-/-‘. Check the uppercase and lowercase again. If the authors want to distinguish genes between human and animal, the format of genes in the manuscript needs to be modified and unified.
6. Line 508, ‘MIR146A’; Line 509, ‘miR146a’. The same problem of Q5.
7. It would be better if circRNAs and lncRNAs can be summarized in table 2 or 3.
8. Add another column of target genes of miRNA in Table 1, and distinguish with miRNA effect.
9. Figure 2 illustrates histone acetylation and deacetylation in OA. What about histone methylation and demethylation in OA ?
10. The format of references should be revised. Like, line 743-IJMS and line 779-International Journal of Molecular Sciences; line 776-Osteoarthritis and Cartilage line 902-Osteoarthritis Cartilage. Check all the references again.
Reviewer 2 Report
This review looked at the epigenetics of OA as a treatment target. The authors narrate the material effectively; however, they might have explained more elaboration.
Abstract: The section will need to be rejiggered. An abstract, as a standard structure, allows readers to rapidly grasp the gist or idea of your article while still following the extensive information, analyses, and arguments. However, the overall writing needs to be more coherent.
The manuscript began with "The concept of epigenetics..." description, which is conscious of today's modern technology. Begin, in my opinion, with the state of the "problem and limitation" associated with the "disease of concern," followed by why breakthroughs in epigenetics study/data/research are required. Modify the section title
Line 79- 94: I encourage the writers to provide diagrams/figures of the overall mechanisms to illustrate them more explicitly for the new researcher.
Line no. 82: "This cytokine can increase the expression of ECM matrix ......metalloproteinases (MMPs) ..such as type II collagen (COL II) and PGs". However, under certain conditions during OA, the enhancement of TGF- β1 and/or TGFβ-RII receptor expression in cells may result in the formation of extracellular matrix proteins. Consistent with TGF-'s stimulation of extracellular matrix protein expression and TGF-'s function in fibrosis. Indeed, COL II, TNF-α, and other cytokines are close functional. Thus, previously should describe the overall mechanisms with the related diagram presentation.
Line 130. The following sentence, "Zhu et al. investigated the decrease of peroxisome..this phenomenon that promotes OA degeneration" most probably needs an in-depth explanation
Line 140: "More recently, the same authors decided .. an essential metabolite in the TCA cycle [28]" How do the authors consider the earlier statement as "most recently" because it has mentioned ref 28 (doi:10.2174/1568007033482788), the publication year 2003? Thus, the authors must include the recent statement.
Figure 1. I don't think that the figure "The role of DNA methylation and demethylation in OA" may complete the mechanisms circle to understanding the process toward the OA
Line 196. "Following the same line" refer to the line that talks about
Line 204: The authors mentioned the "Other studies"; however, references are not included
Include the figure for the "Histone methylation and demethylation in OA" process
It urges the writers to describe section 4.2 adequately. The overall description is lineout with several helpful descriptions
Authors may also include additional information on MiRNA-132, MicroRNA-218-5p, MicroRNA-190, and MicroRNA-165.
Include the graphical presentation of the regulation of miRNA in OA
Reviewer 3 Report
The review drafted by Núñez-Carro et al. is well-structured. Here are some comments to improve the overall quality of the manuscript.
1. It is recommended to describe how the literature search was performed in a systematic way.
2. Only the evidence for micro-RNA was summarized in table. It is recommended to do it for the others.
3. It would be interesting to provide a diagram on all the components of epigenetics and how’s their actions in OA pathogenesis.
